# Information-based Value Iteration Networks for Decision Making Under Uncertainty

**Cynthia Chen**
Allen Institute
Seattle, WA 98109

**Samantha N. Johnson** *
University of Chicago
Chicago, IL 60637

**Cindy Poo**
Allen Institute
Seattle, WA 98109

**Michael A. Buice**
Allen Institute
Seattle, WA 98109

**Koosha Khalvati** †
Allen Institute
Seattle, WA 98109

## Abstract

Deep neural networks that incorporate classic reinforcement learning methods, such as value iteration, into their structure significantly outperform randomly structured networks in learning and generalization. These networks, however, are mostly limited to environments with no or very low uncertainty and do not extend well to partially observable environments. In this paper, we propose a new planning module architecture, the VI$^2$N (Value Iteration with Value of Information Network), that learns to act in novel environments with high perceptual ambiguity. This architecture over-emphasizes reducing uncertainty before exploiting the reward. VI$^2$N can also utilize factorization in environments with mixed observability to decrease the computational complexity of calculating the policy and to facilitate learning. Tested on a range of grid-based navigation tasks, each containing various types of environments with different degrees of observability, our network outperforms other deep architectures. Moreover, VI$^2$N generates interpretable cognitive maps highlighting both rewarding and informative locations. These maps highlight the key states the agent must visit to achieve its goal.

## 1 Introduction

Deep neural networks have provided powerful end-to-end solutions to Reinforcement Learning (RL) problems that map perception to action (François-Lavet et al., 2018). One can approach this end-to-end learning in a classic supervised fashion, especially when provided an expert policy to imitate. However, several studies have shown that incorporating cognitive/classic RL mechanisms, such as simulation of future events and experience replay, improve the learning process significantly. For example, Value Iteration Networks (VINs) incorporate long-term planning (the simulation of future events) by implementing the value iteration algorithm (i.e., a sequence of Bellman updates) via convolutional layers (Bellman, 1957; Tamar et al., 2016; Niu et al., 2018; Zhang et al., 2020; Ishida & Henriques, 2022). Trained either by reward or through imitation of an expert's actions, VINs can learn to navigate in fully observable novel environments significantly better than fully connected and untied convolutional networks (Tamar et al., 2016). Furthermore, their generated model of the environment correctly identifies the rewarding areas (e.g., the goal state).

While VINs and deep reinforcement learning architectures in general have been very successful in many applications, they face a tremendous challenge in many real-world scenarios due to perceptual ambiguity (Ni et al., 2022). Perceptual ambiguity, often called *partial observability*, introduces uncertainty about the current state of the environment. Therefore, the agent must form a probability distribution, or "belief", over its current state and choose its action based on this belief. Even simple networks with a probabilistic belief representation outperform networks with more sophisticated

---

*Co-first author; Work conducted during an internship at the Allen Institute.

†Emails:{cynthia.chen, cindy.poo, michaelb, koosha.khalvati}@alleninstitute.org; snjohnso@uchicago.edu.

encoding and RL modules that perform well in challenging fully-observable environments (Ni et al., 2022; Karkus et al., 2017). However, the main challenge in uncertain environments is that action selection is based on the current belief, not the belief representation. Therefore, more advanced policy/planning modules are required to perform well in more complex uncertain environments.

Formally defined by the Partially Observable Markov Decision Process (POMDP) framework (Thrun et al., 2005), optimal decision making under uncertainty is not achievable in polynomial time (Sondik, 1978). Additionally, powerful sub-optimal approximations involve sampling and tree search techniques with no differentiable implementation, hindering their use in neural network implementations (Ross et al., 2008). Because of these limitations, the current state-of-the-art value iteration network for decision making under partial observability is founded upon a very simple POMDP-solver, QMDP, which assumes that all uncertainty disappears after the first step (Karkus et al., 2017). While this allows for a differential heuristic, this assumption causes the solver to fail in highly uncertain environments.

This paper proposes a new network architecture, the VI$^2$N (Value Iteration with Value of Information Network), that can learn to plan in unseen environments with high uncertainty. VI$^2$N is based on the *Pairwise Heuristic* (Khalvati & Mackworth, 2013), which calculates the solution of sub-problems where the belief state is limited to only a pair of states. Since the Pairwise Heuristic can be calculated by the Bellman equation, it can be implemented with a neural network similar to the VIN. Moreover, VI$^2$N can utilize factorization in mixed observable environments, formally modeled with Mixed Observable Markov Decision Processes (MOMDPs) (Ong et al., 2010; Araya-López et al., 2010). This factorization leads to a huge reduction in computational and memory complexity of the network. We demonstrate the power of our approach by testing it on navigation problems in the presence of different types of uncertainty in various environments. In addition to superior performance in challenging environments with high degree of ambiguity, VI$^2$N generates interpretable maps highlighting key states with high reward and information.

## 2 BACKGROUND

### 2.1 MARKOV DECISION PROCESS (MDP)

Sequential decision making is usually expressed as a Markov Decision Process (MDP) (Thrun et al., 2005). Formally, an MDP is $(S, A, T, R, \gamma)$ where $S$ is the set of states in the environment, $A$ is the set of available actions to the agent, the transition function $T : |S| \times |A| \times |S| \to [0, 1]$ defines $T(s, a, s') = P(s'|s, a)$, the probability of ending up in state $s'$ by performing action $a$ in state $s$, $R : |S| \times |A| \to \mathbb{R}$ is a bounded function determining the reward gained in state $s$, shown as $R(s)$, and $\gamma \in (0, 1]$ is the discount factor for the reward (Thrun et al., 2005). Starting from an initial state, $s_0$, the goal of the agent is to come up with a policy for action selection, $\pi$, that maximizes the total discounted reward. Since the system is Markovian, the policy can be expressed as a mapping from states to actions, i.e. $\pi : |S| \times |A| \to [0, 1]$. The optimal policy $\pi^*$ is $\pi^* = \arg\max_\pi \sum_{t=0}^{H} \gamma^t \mathbb{E}[R(s_t)|\pi, s_0]$ where the horizon $H$ defines the length of this sequence. In classic Reinforcement learning, the value iteration algorithm computes the optimal policy through a series of Bellman updates (Bellman, 1957), i.e. $V_t(s) = \max_a Q_t(s, a)$ where $Q_t(s, a) = \left[ R(s, a) + \gamma \sum_{s' \in S} T(s, a, s') V_{t-1}(s') \right]$ $(t \leq H)$. In deep reinforcement learning, this optimal policy/mapping is learned with a network with the state $S$ (or a representation of it, $\phi(s)$) as the input and the action as the output of the network (François-Lavet et al., 2018).

### 2.2 VALUE ITERATION NETWORK (VIN)

When the transition function is spatially invariant, a neural network can learn $T$ and $R$ by implementing the Bellman equation with convolutional layers (Tamar et al., 2016)). More specifically, given the map of the environment and the current state of the agent (e.g., its position on the map) as the inputs and an expert's action or reward as the output, VIN learns convolutional kernels of $f_R$ and $f_P$ representing reward and transition functions. Such a network with the integration of value iteration as an explicit planning module, generally known as a Value Iteration Network (VINs), significantly outperforms networks with similar computational power (e.g., layers) in learning to plan in unseen environments (Tamar et al., 2016). Originally built for simple lattice worlds with spatially invariant transition functions, value iteration networks have been significantly improved

in terms of applicability to domains with more complex structures over the past years. All of these improvements, however, are still mainly limited to fully observable environments (Niu et al., 2018; Zhang et al., 2020; Ishida & Henriques, 2022).

## 2.3 PARTIALLY OBSERVABLE MARKOV DECISION PROCESS (POMDP)

Existence of uncertainty in the real world, especially in the form of perceptual ambiguity, has made MDPs impractical in many situations (Thrun et al., 2005). Similarly, state-of-the-art networks reaching extraordinary performance in very complicated yet fully observable tasks often fail to handle seemingly small amounts of ambiguity in the environment (Ni et al., 2022). Partially Observable MDPs (POMDPs) represent the closest approach to MDPs that deals with the uncertainty by adding an observation set and observation function to its framework. Formally, a POMDP is a tuple $(S, A, Z, T, O, R, \gamma)$ where $S, A, T, R$, and $\gamma$ are defined very similar to their definition in MDP. $Z$ is the set of observations and $O : |S| \times |Z| \to [0, 1]$ is the observation function determining probability of observation $z$ in state $s$, i.e. $O(s, z) = P(z|s)$. In a POMDP, the agent is not fully aware of its current state. Therefore, it has to maintain a probability distribution over states, often called its *belief* $b(s)$. Starting from a prior probability distribution over states of the environment, called the initial belief ($b_0$), the goal is to maximize the expected discounted reward (Ross et al., 2008). For a POMDP, the optimal decision policy $\pi^*$ can be expressed as a mapping from belief states (probability distributions over states) to distribution of actions that maximizes the total expected reward (Sondik, 1978), i.e. $\pi^* = \arg\max_\pi \sum_{t=0}^{H} \gamma^t E[R(s_t, a_t, z_{t+1}, s_{t+1})|b_t, \pi]$. The uncertainty about the state makes the agent navigate in the belief state space instead of the state space. At time step $t$, the belief state $b_t$ is updated based on the previous belief state $b_{t-1}$ after action $a_{t-1}$ and observation $z_t$ as follows: $b_t(s) \propto P(z_t|s, a_{t-1}) \sum_{s' \in S} P(s|s', a_{t-1}) b_{t-1}(s')$. Partial observability also makes the problem of finding the optimal policy exponentially more complex than the MDP. While the optimal policy of an MDP can be found in polynomial time, finding the optimal policy of a POMDP is NP-hard (Thrun et al., 2005). As a result, the optimal policy can only be approximated by methods such as heuristics, sampling, and search trees (Ross et al., 2008; Khalvati & Mackworth, 2013).

## 2.4 MIXED OBSERVABLE MARKOV DECISION PROCESS (MOMDP)

Although most environments in the natural world incorporate elements of uncertainty, not all aspects of one's current state are unknown. By factorizing the state space into different dimensions, one can separate the visible factors, which contain full observability, from the hidden factors, which are uncertain. An environment with a factorized state space can be referred to as a Mixed Observable Markov Decision Process (MOMDP) (Ong et al., 2010; Araya-López et al., 2010). Although MOMDPs still involve the computation of a belief probability distribution over states, the number of states this belief distribution needs to be computed over can be reduced significantly by only computing solutions for belief distributions on the hidden, or uncertain, factors. Notably, there is not much difference in MOMDP notations other than factorizing the state space into visible/fully observable ($S_v$) and hidden/partially observable ($S_h$), where $S = S_h \times S_v$. Transition, reward, and observation functions could still depend on the whole state. However, in many cases, the factorization could be applied to these functions as well. Notably, not all POMDP solvers can benefit from factorization, but the ones that do can compute equally accurate solutions with less computations and memory.

## 2.5 DEEP NETWORKS FOR SOLVING POMDPS AND MOMDPS

Since POMDPs and MOMDPs have an additional observation function compared to MDPs, deep architectures for decision making under partial observability represent observation kernel $f_Z$ in addition to transition function kernel $f_P$ and reward kernel $f_R$. Notably, the structure of these kernels depend on the problem, but the general methodology and algorithm of learning is the same. For example, in a navigation problem, the agent might observe only its surrounding instead of the whole world, which makes the observation kernel a $3 \times 3$ convolutional kernel. In a different domain, observation kernel might consist of only a $1 \times 1$ convolution over the agent's current position. Similar to classic POMDP solvers, a POMDP solver network consists of two modules of belief update and policy, forming a recurrent architecture together. The belief update can be easily implemented in a network, as exemplified in the QMDP-Net architecture (Karkus et al., 2017) (Also

demonstrated in figure A–1). However, designing a powerful policy module is very challenging due to the differentiability requirement. As a result, current networks for solving POMDPs have a very simple policy module, e.g., a model-free RL module (Ni et al., 2022) and QMDP. MOMDP solver network has the same structure, but with a smaller-sized belief state space covering only the unobservable factor of the state space. Like classic POMDP solvers, how the policy module takes advantage of mixed observability is significantly more important than the belief update module.

## 3 MODEL

Our main goal is to provide a better policy module for the value iteration networks in partially observable environments. Our architecture is founded upon a "Pairwise Heuristic". Originating from Bayesian active learning in which the heuristic is used to find the correct hypothesis with a set of noisy tests (Golovin et al., 2010), the Pairwise Heuristic has also been used in a general-purpose POMDP-solver when the environment model is fully known (Khalvati & Mackworth, 2013). Here we present this version, slightly modified for our framework.

### 3.1 THE PAIRWISE HEURISTIC FOR SOLVING POMDPS

The main idea of the pairwise heuristic is to use solutions of the smallest sub-problems that still consider the uncertainty about the true hypothesis/state, which would be pairs (sets of 2) of hypotheses/states (Golovin et al., 2010). In a POMDP, this would be the set of $|S|(|S| - 1)/2$ optimal policies in each of which the belief is .5 for two states. The expected total reward of each of these policies is the *value of the pair*, shown by $V(s, s')$, for $s, s' \in S$. This is differentiated from $V(s)$, which corresponds to the value of a state based on the optimal policy computed through Value Iteration on a fully observable representation of the state space. Calculating pairwise optimal policies is still computationally very expensive. Therefore, the pairwise heuristic for POMDPs applies an additional heuristic to calculate $V(s, s')$ (Khalvati & Mackworth, 2013). For each pair, it resolves the uncertainty first and then exploits the reward. Given the observation function, the uncertainty is already resolved for some pairs of states. To be more precise, it is highly unlikely to have a notable probability/belief for states with different observations. These pairs are "distinguishable". For other pairs of states, i.e. indistinguishable ones, the Pairwise Heuristic resolves the uncertainty by going to distinguishable pairs. Two states are distinguishable if there is a high probability that different observations are recorded in the two states. Formally, $s$ and $s'$ are distinguishable if and only if:

$$\sum_o p(o|s)(1 - p(o|s')) + p(o|s')(1 - p(o|s)) \geq 2\lambda \tag{1}$$

$\lambda$ is a constant that is specified by a domain expert. If there is no noise in observations, this value is 1. Otherwise, this value is close to but less than 1. The pairwise value ($V(s, s')$) of distinguishable pairs is simply the average of the value function of each of the states in the underlying MDP model of the environment (assuming full observability in the environment), i.e., $.5(V(s) + V(s'))$. To find the value function of the indistinguishable pairs, we use a value iteration algorithm in an MDP where the states are pairs of states of our original problem. The transition function of this MDP is determined by the joint transition probability distribution of the original environment:

$$T((s, s'), a, (s'', s''')) = p((s'', s''')|(s, s'), a) = p(s''|s, a)p(s'''|s', a) \tag{2}$$

The reward of each pair is simply the average reward of the two states in the original problem:

$$R(s, s') = 0.5(R(s) + R(s')) \tag{3}$$

Therefore, the Bellman equation of our pairwise value iteration algorithm is as follows:

$$V_k(s, s') = max_a[R(s, s') + \gamma \sum_{s'', s'''} T((s, s'), a, (s'', s'''))V_{k-1}(s'', s''')] \tag{4}$$

Initial pairwise values, i.e., $V_0(s, s')$, in the above equation is $.5(V(s) + V(s'))$ for distinguished pairs and the minimum possible reward for indistinguished ones. To select an action, the Pairwise

Heuristic POMDP-solver maximizes the expected value of pairs using the joint belief state, i.e., $b(s, s') = b(s)b(s')$. This essentially represents Value Iteration over the set of pairs of beliefs.

$$a_k^* = \arg\max_a \sum_{(s,s')} b(s, s')Q((s, s'), a) \tag{5}$$

If the probabilities of all states, except the most likely one, become negligible, the selected action would be the optimal action of the underlying MDP for that most likely state.

Notably, in a MOMDP, any pair of states with different visible factors are distinguishable. Therefore, the heavy computation of value iterations on pairs would be reduced to $|S_v|.|S_h|.(|S_h| - 1)/2$, which is a significant decrease in computational complexity when $|S_h| << |S_v|$. More specifically these pairwise values can be represented as $V(s_v, s_h, s_h')$ where $s_v \in S_v$ and $s_h, s_h' \in S_h$.

## 3.2 $VI^2N$ ARCHITECTURE

All of the pairwise heuristic POMDP solver processes have a straightforward differentiable implementation. The central part of this solver is the pairwise value iteration (Eq. 4), which uses the pairwise transition (Eq. 2), and reward functions (Eq. 3). Moreover, the initial pairwise values are determined by the value of states ($V(s)$) in the underlying MDP and the distinguishability of each pair of states (Eq. 1). The network implementation of these components is demonstrated in figure 1a.

Starting from the value iteration algorithm implemented by a VI module, the network learns $f_P$ and $f_R$, determining $T(s, a, s')$ and $R(s)$ of the environment. They are combined as the value iteration equation dictates, to yield $V(s)$. From this point, the objective becomes converting elements of the environment to a pair-space representation to allow for the $VI^2$ module implementation. Specifically, we must convert $T(s, a)$ and $R(s)$ into $T((s, s'), a)$ and $R(s, s')$ for all $s, s' \in S \times S$. We can convert $R(s)$ using an averaging layer across all $s$. As mentioned before, while the principles of value iteration networks remain the same, the details of kernels depend on the problem. Here, following the previous works we focus on 2D goal-based navigation with different types of uncertainty (More details in section 4), where the agent moves to one of the adjacent cells. Therefore, the transition function is implemented by a $3 \times 3$ kernel. This kernel must be transformed into a transition kernel for the pairwise state space, which involves increasing the size of the kernel from $(3, 3)$ to $(2(\sqrt{S} + 1) + 1, 2(\sqrt{S} + 1) + 1)$ to allow for row and column transitions between pairs (assuming the grid world is a $\sqrt{S} \times \sqrt{S}$ square). This kernel is constructed using the learned transition probabilities from the VI Module and has a number of channels equal to the number of actions available in the environment similar to $f_P$. All nine values of each channel of $f_P$ would be mapped to the main diagonal of the pairwise transition kernel in the corresponding channel as demonstrated in figure 1b.

We must also determine which set of pairs $(s, s')$ are distinguishable. We implement this by applying the convolutional kernel for the observation function, $f_Z$, to all states (the gridworld map) to get matrix $Z$. Then we use the outer product of $Z$ and $1 - Z$ and compare it with a threshold to implement Eq. 1. We express the distinguishability by a binary $|S| \times |S|$ matrix, $D$. The pairwise value initialization ($V_0(s, s')$) is done using matrix multiplication of $D$ and $.5(V(s) + V(s'))$ (for distinguished pairs) in addition to multiplication of $(1 - D)$ and $min_{R(S)}$ in the shape of an $|S| \times |S|$ matrix (for indistinguishable pairs). With the pairwise reward and transition function calculated, pairwise value iteration (Eq. 4) is just another VI module, which we call the $VI^2$ module. Finally, the action selection (Eq. 5) is done by multiplying the pairwise belief state (outer product of belief by itself) with pairwise Q values and applying the max pooling layer (figure 1b).

The size of $VI^2N$ decreases significantly if the state space can be factorized into visible and hidden factors: $s = (s_v, s_h)$ where $S = S_V \times S_h$. In this case, the pairwise modules of $R$, $V$, and $D$ would be $|S_v| \times |S_h| \times |S_h|$ (with size = $|S_v|.|S_h|^2$) instead of $|S| \times |S|$ (with size = $|S_v|^2.|S_h|^2$) as demonstrated in figure 1d. This is a significant reduction when $|S_h| << |S_v|$.

## 4 RESULTS

We compared $VI^2N$ with the QMDP-Net on two types of grid-world navigation problems with different types of uncertainty and challenges. Since QMDP-Net has been shown to perform significantly better than unconstrained networks (Karkus et al., 2017), we did not include them in our analysis.

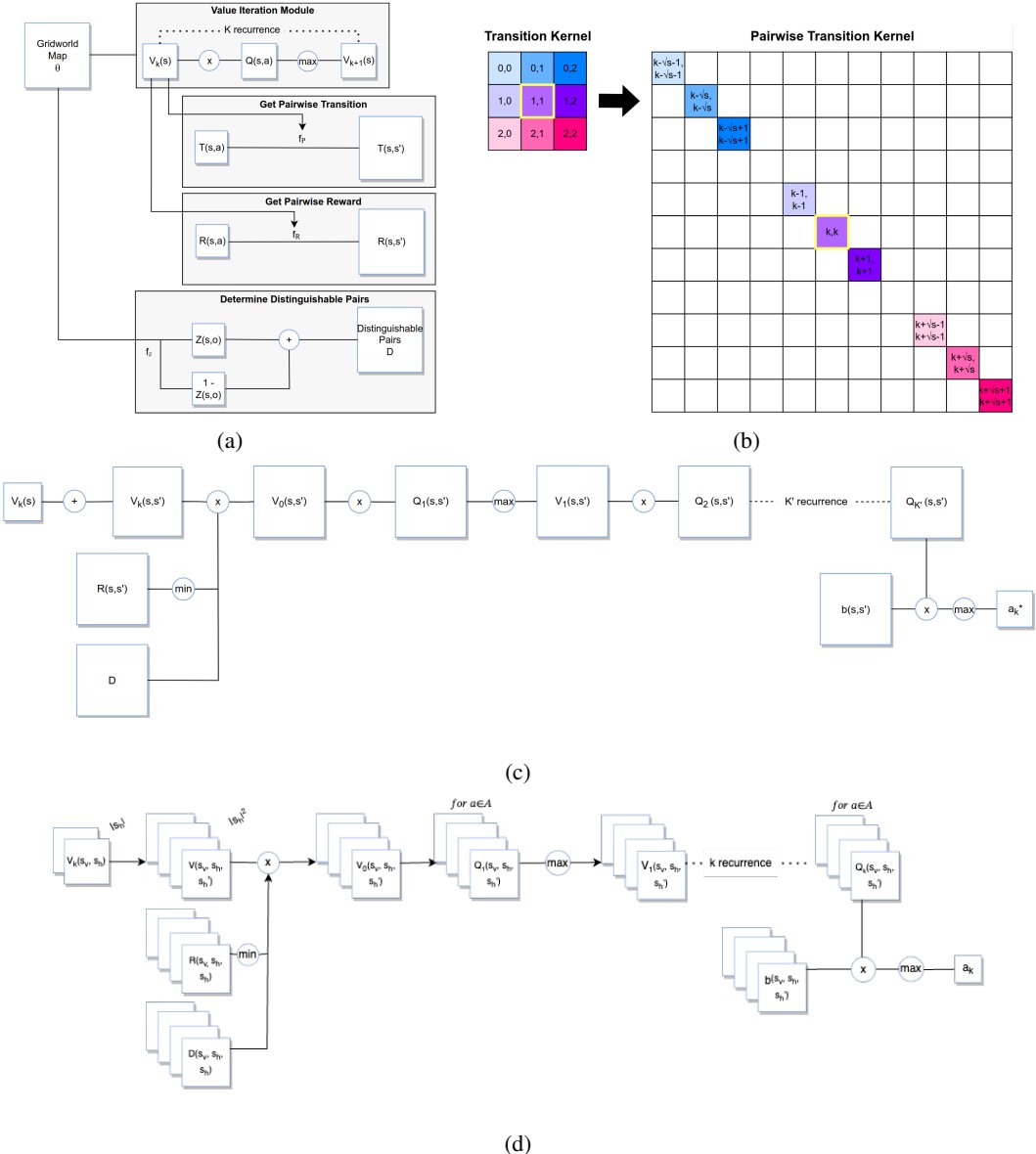

Figure 1: **VI²N Architecture**. **a)** Required preprocessing to prepare for Pairwise Value Iteration **b)** Example conversion of one channel of the transition kernel $f_P$ to the corresponding channel of pairwise transition kernel. Example assumes a 4x4 environment of 16 states, where $S =$ number of states, kernel size $= 2(\sqrt{s}+1)+1$, and $k = \lfloor \text{kernel size}/2 \rfloor = \sqrt{s}+1$. **c)** Pairwise Value Iteration modules, which uses outputs from part (a) to determine the selected action. **d)** Factorized Pairwise Value Iteration modules (Factorized version of part (c)).

Additionally, we tested a state-of-the-art transformer-based architecture for solving fully observable MDPs on our task (Chen et al., 2021). That model failed to succeed on over 10% of our test cases, so we excluded it from our analysis. We used the same belief update mechanism for all agents to have a fair comparison between the policy modules. Moreover, we set the number of recurrences in the VI module of QMDP-net equal to the total number of recurrences in VI and VI$^2$ modules of our network. Importantly, the transition kernel did not get updated in the pairwise module (VI$^2$) (we did not pass gradients in this module). Therefore, Q-MDP net did actually have a computational advantage over VI$^2$N in terms of the number of free parameters. Although QMDP is computationally advantageous to VI$^2$N, the increase in success rate compensates for that increased complexity.

Our first task/problem set was the standard single-goal navigation where the position of the goal is visible in the input map, but the agent does not know its own position. Instead, it observes its surroundings. This problem set included several environments with various amounts of perceptual ambiguity. In the second navigation task/problem set, the agent knows its position but is unaware of the exact position of the goal. Instead, a few *possible* locations for the goal are given to the agent. The agent should go to a "landmark" spot to find the goal. Multiple potential goal positions make the state space significantly larger. However, the state space can be represented as (agent's position, goal's position) in which the first factor is fully observable. Each of the two tasks/problem sets includes different structural versions, 4 in the first task and 3 in the second. Networks are trained on each structure (15000 to 30000 instances), and tested on novel environments of the same structure (3000 to 6000 instances). Details of the process can be found in appendix A.

## 4.1 TASK 1 - OBSERVABLE GOAL, UNKNOWN POSITION

For the first task, we designed four environments with different levels of perceptual ambiguity. Each type of environment was tested at various levels of obstacle density. The observation and action functions were consistent among the environments to have a systematic comparison in terms of uncertainty and complexity of the decision-making. The actions were "right", "up", "down", and "left", moving the agent one cell in the direction specified by the name, and also action "stay". The agent was able to observe the cell it was on and also the neighboring cells in each of the cardinal directions. Since our focus was on perceptual ambiguity, not handling noise in sensors and actuators, both of our sensors and actuators were noise-free. Notably, we tested the networks on environments with noise, and both of them maintained their performance.

We started with a "random" environment. In the random environment, obstacles are randomly placed within the arena at both 5% and 10% density/sparsity levels (figure 2a, top left). With an average of 20 or 40 obstacles in this environment, uncertainty would be resolved in a few steps. As a result, both networks had a very high success rate (table 1, top row). We increased the ambiguity by adding the constraint of minimal continuity in each axis to the random environment, which produces very few blocks with a side size of 4. This type of environment called "blocks", was generated to model environments where obstacles are randomly placed as independent clusters ((figure 2a, bottom left). With the increase in perceptual ambiguity compared to the "random" environment, both networks' performance dropped. However, the drop was lower for VI$^2$N (table 1, the second row from the top). Our third environment called "walls", contained long walls parallel to the border in an empty arena, resembling long hallways for robots with sonar sensors ((figure 2a, top right). The superiority of VI$^2$N became appreciable in this challenging environment, where the middle walls and borders are not easily distinguishable (table 1, the second row from the bottom).

Our most challenging environment was called "symmetric". In this environment, four copies of a smaller random environment are placed in each corner of a larger grid-world (figure 2a, bottom right). The density of each of the four repeated environment blocks was 5%, 10%, or 15%. This environment requires more long-term planning and information gathering, as it has more indistinguishable states that could lead to incorrect assumptions about belief. In this environment, the VI$^2$N drastically outperformed the QMDP-Net, which demonstrates its ability to use long-term planning to generate effective policies. Moreover, VI$^2$N was more robust to changes in sparsity in complex environments, whereas the QMDP-Net performance dropped at a higher rate as each type of environment became more challenging with increased sparsity. We also tested the robustness of networks against minor changes in the new environment by corrupting the symmetry. We corrupted the full symmetry of our 10% density test set by changing five random empty cells to obstacles in one of the rooms (5% of that $10 \times 10$ room). The median success rate dropped to 68% for VI$^2$N, and 48% for QMDP-net.

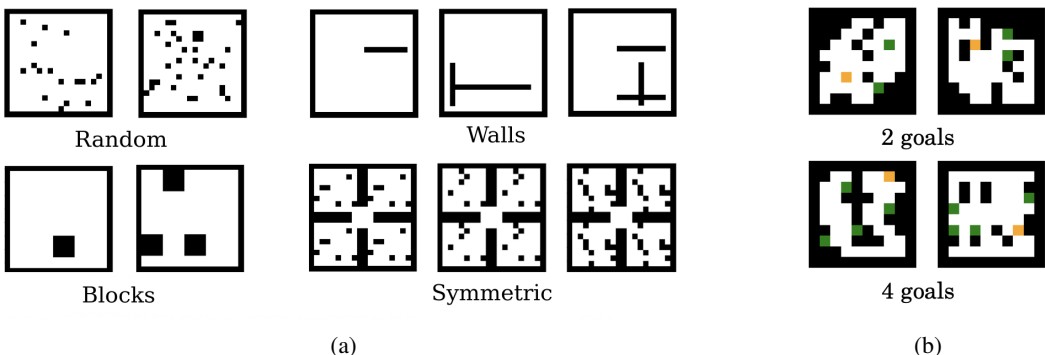

Figure 2: **Example testing environments**. **a)** For the observable goal, unknown position task, there are four types of random, blocks, walls, and symmetric environments, with various density rates. The goal is not marked in those maps. **b)** For the unknown goal, observable position task, there are different numbers of potential places for the goal in the environment, and displayed are 2-goal and 4-goal examples. The potential goals are shown in green, and the landmark is shown in orange. The agent finds out where the goal is actually placed if and only if it goes to the landmark spot.

Table 1: Success rate of network solvers in task 1 (unknown agent's position) in different environments.

| Model | Environment | | | | |
|---|---|---|---|---|---|
| | Random(5%) | Random(10%) | Walls(1) | Walls(2) | Walls(3) |
| VI$^2$N | $93 \pm 1\%$ | $95 \pm 1\%$ | $77 \pm 1\%$ | $83 \pm 1\%$ | $82 \pm 1\%$ |
| QMDP-Net | $93 \pm 1\%$ | $96 \pm 1\%$ | $69 \pm 1\%$ | $78 \pm 2\%$ | $80 \pm 2\%$ |
| | Blocks(5%) | Blocks(10%) | Symm(5%) | Symm(10%) | Symm(15%) |
| VI$^2$N | $91\% \pm 1$ | $91 \pm 1\%$ | $76 \pm 3\%$ | $74 \pm 3\%$ | $65 \pm 4\%$ |
| QMDP-Net | $88 \pm 1\%$ | $89 \pm 1\%$ | $61 \pm 3\%$ | $51 \pm 5\%$ | $41 \pm 4\%$ |

## 4.2 TASK 2 - UNKNOWN GOAL, OBSERVABLE POSITION

Our gridworlds for the second navigation task/problem set were generated with a similar script as the "random" worlds used in the first problem set, with the side of each world, $N$, being 10, and an obstacle density of $20\%$. However, we manipulated the number of possible goal locations displayed on the map, $|G|$, for these worlds, increasing the uncertainty of the environment by adding more possible goal locations. Each location added increases the state space by $N \times N$, yielding $|S| = N \times N \times |G|$ states. Our worlds all included a "landmark" location, which produced an observation that revealed the hidden location of the goal to the agent (number of observations $|O|$ equals $|G| + 1$, representing each goal and a "neutral" observation). Other locations had absolutely no information about the position of the goal. Example environments with 2 and 4 possible goal locations are shown in figure 2b.

The potential goal locations either yield a large reward or a large penalty, disincentivizing the agent from going to potential goal places one by one, skipping the "landmark". This is similar to a natural task where an agent knows where it is, but does not know where the reward is. In those cases, the agent may navigate the environment in order to gain information about the goal before proceeding to exploit the reward. This task also explores the concept of a MOMDP, incorporating a visible and hidden component. In this task the visible factor of the state, $S_v$, is the agent's location. The hidden factor, $S_h$, is the goal's position, which is one of the potential locations ($|S_h| = |G|$). Using this factorization, our pairwise modules were 100 times smaller ($N \times N \times |G|^2$ instead of $(N \times N)^2 \times |G|^2$ where $N = 10$). As seen in table 2, our network significantly outperformed QMDP-net in all conditions. Upon further examination of each solver's trajectory, we found that $VI^2N$ visited the landmark state more frequently than QMDP-net, visiting the landmark an average of $96\%$ of the time, whereas the QMDP-net only visited the landmark an average of $34\%$ of the time.

Table 2: Success rate of network solvers in task 2 (unknown goal, observable position) in different conditions (number of potential goal locations and stochasticity of the environment).

| Number of goals | 2 | 3 | 4 | 2 | 3 | 4 |
|---|---|---|---|---|---|---|
| Transition Function | Deterministic | | | Stochastic | | |
| $VI^2N$ | $98 \pm 2\%$ | $96 \pm 2\%$ | $95 \pm 2\%$ | $91 \pm 2\%$ | $91 \pm 3\%$ | $91 \pm 2\%$ |
| QMDP-net | $57 \pm 4\%$ | $53 \pm 2\%$ | $51 \pm 2\%$ | $27 \pm 2\%$ | $28 \pm 2\%$ | $30 \pm 17\%$ |

Table 3: Success Rate of network solvers in task 2 (unknown goal, observable position) in a $20 \times 20$ environment with 4 goals.

| Model | Success % |
|---|---|
| $VI^2N$ | $79 \pm 3\%$ |
| QMDP-net | $27 \pm 2\%$ |

Table 4: Effect of number of recurrence on VI$^2$N success rate (task 2, $|G| = 4$).

| $k_{VI}$ | $k_{VI^2}$ | Success % |
|---|---|---|
| 5 | 5 | $58 \pm 16\%$ |
| 20 | 1 | $41 \pm 2\%$ |
| 1 | 60 | $45 \pm 9\%$ |
| 40 | 20 | $95 \pm 2\%$ |
| 60 | 40 | $94 \pm 2\%$ |

We also tested the models' robustness to noise by implementing a stochastic transition function. Specifically, we created environments in which the probability of taking the intended action was $60\%$, whereas the probability of moving left or right relative to the intended action was $10\%$, and the probability of not moving was $20\%$. The entire evaluation process remained unchanged. As shown in Table 2, the gap between the performance of $VI^2N$ and QMDP-net increased. Finally, we tested model scalability in a $20 \times 20$ deterministic environment with 4 goals, where $VI^2N$ found the goal $79\%$ in average, while QMDP-net's average success rate was $27\%$ (Table 3).

### 4.3 EFFECT OF RECURRENCE

$VI^2N$ has two planning modules, each with a recurrence parameter $k$ ($k_{VI}$ and $k_{VI^2}$) specifying the planning horizon. To assess the importance of these modules and their associated parameters, we trained and tested $VI^2N$ with different recurrence parameters in the task 2 with 4 potential places for the goal. As demonstrated in table 4, short horizons hurt the performance, while horizons above certain values do not lead to any improvement, similar to classic value iteration. Moreover, the better performance of a network with $k_{VI} = k_{VI^2} = 5$ compared to networks with $k = 1$ in one of the modules as well as QMDP-Net highlights the importance of both planning modules in $VI^2N$, one for reward exploitation, and the other for resolving uncertainty.

### 4.4 INTERPRETABILITY

Besides the superiority of VI$^2$N, measured by an objective measure of success rate, our method produces interpretable maps that highlights both informative and rewarding areas. Specifically, in addition to producing value maps representing the space in terms of the value function of single states ($V(s)$), VI$^2$N specifies informative areas via marginal pairwise values, i.e., $\sum_s V(s, s')$, as demonstrated in figure 3. This map significantly contributes to the interpretability of our method, explaining why specific actions were performed and why certain areas were visited more, especially when they were not directly related to the source of reward (goal). Informative states are not represented in the QMDP-net value function, even in the environments where the QMDP-Net performs well. This is expected, as the algorithm behind policy generation of QMDP-Net does not take resolving uncertainty into account.

### 5 DISCUSSION

We have introduced the VI$^2$N as a deep learning architecture for decision making under uncertainty, modeled after the fully differentiable Pairwise Heuristic. The VI$^2$N architecture demonstrates the ability for long-term planning for resolving the uncertainty which exceeds the capacity of previously

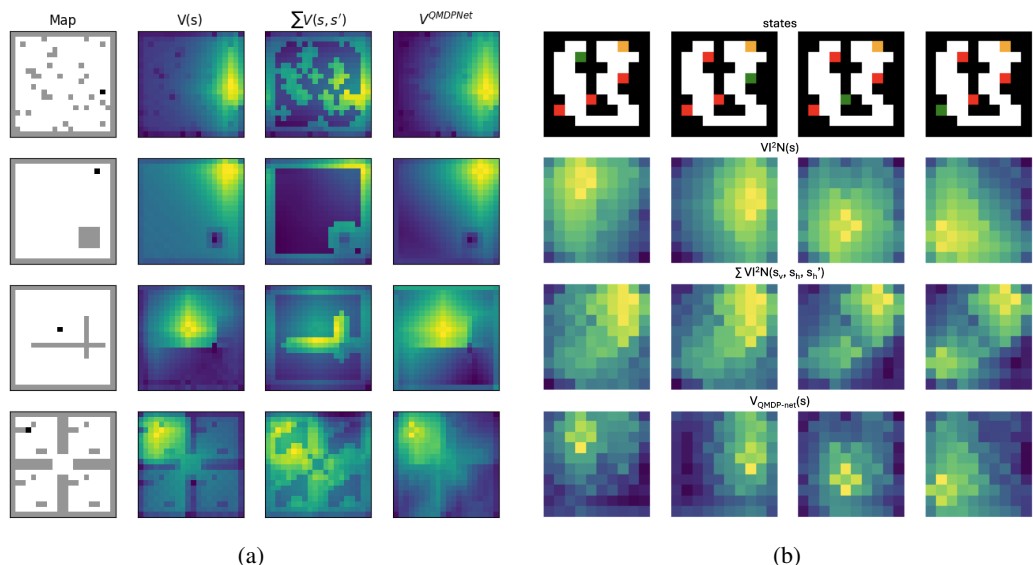

Figure 3: **VI$^2$N represents rewarding areas through the value function of single states and informative areas through the value function of pairs. QMDP-Net focuses only on the reward. a)** Example generated maps in the first problem set. The leftmost maps represent the environment where the gray areas are obstacles and the black cell represents the goal. **b)** Example generated maps in the second data set with for goal spots. Rewarding goal is shown in green, the misleading "goals" are shown in red, the landmark state is orange and the obstacles are shown in black

proposed network architectures seen in the VIN and the QMDP-Net, especially in challenging environments with high perceptual ambiguity. Moreover, it's the only network that can take advantage of mixed observability of the environment and factorization. Furthermore, in addition to *reward value* maps, VI$^2$ generates *information value* maps, highlighting the informative areas in respect to the reward (goal). Since the main focus of our work is on the planning/policy module of the network, our environments were simple 2D binary grid worlds similar to VIN and QMDP-Net papers. We expect improvements of classic VIN over the past years (Niu et al., 2018; Zhang et al., 2020; Ishida & Henriques, 2022) to be easily applicable to our network as the main component of our network is still a VI module. In fact, applying these improvements is an exciting future research direction to extend the applicability of our network.

Similar to other networks in the VIN family, the most significant limitation of VI$^2$N is the requirement of spatial invariance for transition and observation functions. Moreover, all of the tasks in this paper were 2D navigation. This allowed us to have a systematic comparison based on different types and levels of uncertainty. Testing other tasks, such as grasping, would definitely contribute to the reliability of our results. However, designing scalable, challenging, and intuitive setups for other tasks is unfortunately complicated. For example, as shown in the QMDP-Net paper, the available grasping environment is not even challenging for the classic QMDP algorithm with more than 98 percent success rate (Karkus et al., 2017). It is also worth mentioning that pairwise modules are easily scalable for environments with higher dimensions. For example, the only major structural change for a 3D navigation task is using a larger kernel for pairwise transition function. Finally, our results are limited to learning from an expert (and not from reward reinforcement).

## 6 REPRODUCIBILITY STATEMENT

Details of the experiments, including network parameters, the training and testing processes, and environment generation, are available in the appendix A. The code for implementing the networks and generating environments is available at https://github.com/cynthiachennn/vi2n.

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

## A  NETWORK ARCHITECTURE AND DETAILS OF EXPERIMENT

A value iteration network for decision-making in partially observable environments is a recurrent network consisting of two main modules: The belief update (Bayes Filter) module and the policy/planning module (figure A–1, left). The input to the network is mainly the map of the environment. After selecting each action (output), the network also receives an observation which is used along with the chosen action to update the belief for the next actions. Belief update operations, such as update after action and normalization, are easily implementable in a network, as exemplified in the QMDP-Net paper Karkus et al. (2017)(figure A–1, right). The policy module selects the action given the map and the belief, as demonstrated in figure 1 of the main text.

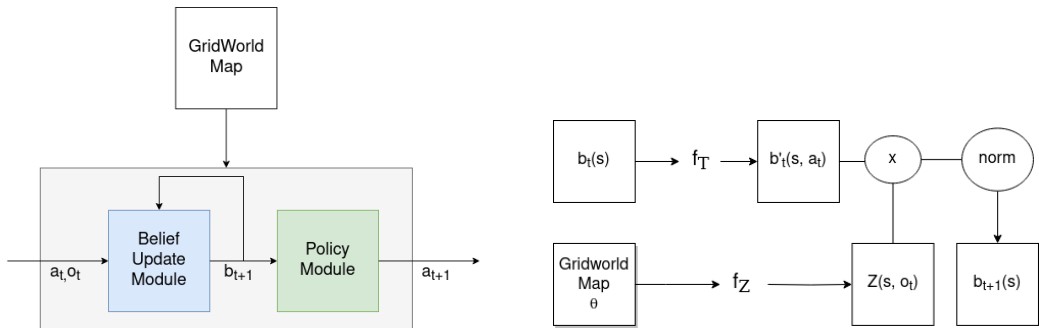

Figure A–1: Value iteration networks for decision making under partial observability. a) The network consists of two main modules: the belief update module and the policy module. b) Belief update module: This module implements the step of belief update in the POMDP as explained in the main text and other works such as QMDP-Net Karkus et al. (2017).

For the first task, the input map is two binary matrices indicating obstacles v.s. empty space, and goal v.s. no goal, respectively. This means that, given the size of the map in all of our environments ($20 \times 20$), this map was a binary $2 \times 20 \times 20$ tensor. As mentioned in the main text, the actions were 'right', 'up', 'down', and 'left', moving the agent one cell in the direction specified by the name and also action 'stay'. The observation range was also 1, represented by a $3 \times 3$ kernel. The agent was able to observe the neighboring cells in each of the cardinal directions. Moreover, the goal state was observable when the agent was in it. Therefore, there were 17 observations in total. As mentioned in the main part of the paper, the action and observation functions were noise-free / deterministic. Therefore, and for simplicity, we represented the observation function as a $5 \times 3 \times 3$ tensor, using five channels instead of 17, each of which is representative of one of the 5 (out of 9) observable cells.

As our goal was to compare the policy modules, we assumed that the agent knows the observation function and updates the belief perfectly in all tested architectures (including QMDP nets). Therefore, each data point consisted of the grid world map, the belief as the input, and the selected action as the output. Networks for all environment types were trained and tested within the same type except "random" and "blocks". The training set for these two was simply the aggregate of data points of both types (we basically combined data points of these environments during training).

Each training data set for the first problem consisted of around 12000 environments, each of which with two or three different start states. For each type, density rates were distributed uniformly, e.g., 6000 environments with $5\%$ density for the "random" and "blocks". The train-validation ratio was $95\% - 5\%$. Moreover, the test success rate was calculated by running the generated model on 1,000 novel environments of the same type, each of which with 20 different start states for each density. The batch size was 100 for all methods and environments. The maximum number of epochs was 1000 for QMDP-Net and 300 for VI$^2$N as it was computationally more expensive. Both networks reached a stable point in the last quarter of training, suggesting that more epochs were unnecessary. All implementations (VI$^2$N and QMDP-Net) were in Pytorch, and the experiments were done with GPU boxes with 4 NVIDIA GeForce RTX 2080 Ti graphic cards. Each epoch took at most 1 hour (for VI$^2$N).

For our second task, the input map consists of obstacle, goal, and landmark channels. The obstacle channel is a binary matrix like the obstacle matrix for our first problem set, representing obstacles vs.

empty space. Each possible goal location in the environment was represented in its own layer, as the existence of one goal means the other goals are not valid in that state. Finally, the landmark, which returns an observation revealing the true goal, was represented by another binary matrix indicating landmark state vs non-landmark state in the last layer. All of the grid worlds were $10 \times 10$ matrices, with a varying number of possible goals, yielding input maps of size $10 \times 10 \times (2 + |G|)$. In this case, the agent always knew its $(x, y)$ dimension location, so the observation function was represented by a $|G| + 1$ vector, with a "null" observation, as well as an individual observation corresponding to each state. As with the first problem, we first assume the agent knows the observation and transition functions and updates the belief perfectly in all tested functions.

Our training set for the second task is slightly smaller, consisting of around 3000 environments with 5 start states in each environment, totaling about 15000 unique action labels. We used a smaller dataset because of the smaller belief distribution in these tasks. Our test set was also slightly smaller, consisting of 2000 environments with 5 start states for each environment. Like with the first task, the train-validation ratio was $95 - 5\%$, and we used a batch size of 100 and allowed our models to learn for 300 epochs.

