# OpenReview forum: "Information-based Value Iteration Networks for Decision Making Under Uncertainty"
_ICLR.cc/2026/Conference — ICLR 2026 Poster_

### Official Review · Reviewer_bA3z · 2025-10-17

**Soundness:** 2
**Presentation:** 2
**Contribution:** 2
**Rating:** 4
**Confidence:** 2

**Summary:**

The paper introduces a new DNN architecture for RL. This architecture builds on the idea of Value Iteration Networks, extending it to better contend with the partial observability of POMDPs using the idea of Pairwise Heuristic.

**Strengths:**

1. The authors are very clear and direct about the limitations of the paper, which I greatly appreciate.
2. I found the presentation for the most part very clear.
3. The idea seems very natural for the purpose of applying VI nets to POMDPs.

**Weaknesses:**

1. Limited evaluation: only navigation tasks. No RL experiments - only learning from an expert.
2. Limited novelty: the paper takes two developed ideas (Value Iteration Networks (VIN) and Pairwise Heuristic (PH)) and combines them in an apparently straight forward manner.
3. HP tuning: The HP tuning process for the method and the baseline QMDP are (I believe) unspecified. I.e. was there any? Was it for both methods? Was it under equivalent conditions (i.e. same amount of compute dedicated to the tuning of each?) and motivation for the decisions are missing.
4. Statistical significance: There is a measure of stat. signif. in the tables. Is it standard div.? SEM? Other? I believe it is unspecified (unless I've missed it). Is it over different seeds (how many)? Is there a reason seeds are not necessary here? (random init of the DNN seems to justify different seeds, to me).
5. Introduction to an RL audience could be expanded (see questions / comments). Specifically, I would have liked a brief overview of the training process (dataset with expert actions? learning through interaction with the environment? other?). If the authors could please include a brief explanation in the rebuttal, and add a description into the main paper.

**Questions:**

I'm open to changing the review score/s. Specifically:
1. If the authors could motivate well that: the evaluation is sufficient for the method to convincingly dominate the baseline in a major set of tasks. Alternatively, increasing the number and types of tasks in the evaluation.
2. If the authors could motivate well that the combination of VIN and PH is not straightforward.
3. If the authors will add and motivate the HP tuning proces and stat. signif. evaluation.

Additional comments:
1. Uncertainty can refer to different things, that are traditionally dealt with very differently, in RL: the partial observability of POMDPs, the epistemic-uncertainty that drives exploration in sparse-reward domains, and the stochasticity of reward / transitions ("aleatoric" uncertainty) that makes everything more challenging. Although I think the presentation is rather clear in its focus on POMDPs exclusively, I would have liked it to be even more explicit - preferably as early as possibly (i.e. intro), on the types of uncertainty that will / will not be addressed in this work.
2. Lines 019-020: "Tested on a diverse set of domains". Since the evaluation is limited to navigation tasks in grid envs. with goal / position known / unknown, I do not think that this can be considered "a diverse set of domains". I would be more comfortable with a claim along the lines of "a range of grid-based navigation tasks".
3. I would have liked the QMDP baseline to be presented in more detail (perhaps in a related work section?), so that the reader can better understand and contrast the method and the (only, previously SOTA?) baseline.
4. In Section 3 there are multiple references to "the value V". It's not clear to me whether this refers specifically the value of the optimal policy $V^*$, the value of a general policy $V^\pi$, or specifically the value of some expert policy $V^{\pi_e}$. Could the authors specify (And add to the paper)?
5. Table 2 and Table 3 seem to measure the same thing (success rate) in different units (% and out of 1). Have I misunderstood? If not - is there a reason they are not presented in the same units?
6. Non-VI-based baselines for POMDP solving (i.e. standard popular algorithms with an LSTM) would put the results in better context.
7. Can the authors include additional results / discussion of compute cost contrast between QMDP and their method (comparable / one is significantly more expensive than the other?)

Additional  minor comments
1. Line 011: In my opinion (/understanding), value iteration is a dynamic programming method, not an RL method (it relies on knowing the full model of the env., not on learning from a dataset of interactions). I do not mean to nitpick, it is more that it would have been easier for me to follow the narrative had DP rather than RL been the term used.
2. Line 017: "This architecture over-emphasizes..". Over means "too much" -> bad. Is that the intention? Perhaps simply "emphasizes"?
3. Equation 1 sums across s' s', probably should be across s' s. I'd also denote the set $o \in Z$ for the first sum, to improve clarity.

---

> ### Author Response · Authors · 2025-12-04
>
> We thank the reviewer for their time and effort in reviewing out paper. We have addressed W1 and W3 in our Author’s comment, as well as comments 1 & 6.
>
> W2- **Limited novelty from the combination of PH and VI:** The implementation of PH in VIN is novel due to the implementation of the pairwise transition, reward, and value matrices used to calculate the optimal action. In order to utilize PH, we needed to design new transition and value matrices in order to account for the pairwise combination of states, as detailed in Section 3, subheading “VI2N Architecture”. One way to design these matrices would have been to create pairwise combinations of the original $T()$ and $R()$ matrices. However, that would square the number of computations necessary. We implement a custom kernel to create pairwise transition matrices and reward matrices while maintaining efficiency. For our first problem set, our custom transition kernel involves taking advantage of the spatial invariance of the transition function. Our second problem set takes advantage of the factorizable nature of the problem to reduce the size of our pairwise transition kernel and pairwise value representation.
>
> Comments 1, 2, 3, 5 – Our paper has been updated to address these issues.
>
> Comment 4 - **Clearer definition of “the value V”:** $V(s)$ generally refers to the value of a state based on the optimal policy computed through Value Iteration on a fully observable representation of the state space. Pairwise value, $V(s,s’)$, generally refers to the best calculated value of pairs based on our pairwise heuristic algorithm. We have added this clarification to our paper.
>
> Comment 7 - **Compute cost:** We discussed the computational complexity of each solution in section 2 and 3, when describing each model. We mentioned that QMDP is linear and Pairwise Heuristic is quadratic in unfactorized problems. For factorized problems, Pairwise Heuristic is linear in the observable parts. While VI$^2$N is more computationally demanding than QMDP-net, we believe that the increase in success rate given by VI$^2$N compensates for that increased computational demand. We have added a direct comparison to our paper to emphasize this difference.

---

### Official Review · Reviewer_SnV9 · 2025-10-27

**Soundness:** 2
**Presentation:** 1
**Contribution:** 2
**Rating:** 2
**Confidence:** 2

**Summary:**

The paper seems to extend partially observable VIN, in particular the method QMDP-Net, with a pairwise heuristic for solving POMDP. Experiments on simple gridworld tasks show their approach solves more randomly generated tasks than QMDP-Net.

**Strengths:**

The pairwise approach seems to be especially well suited for tasks where the state space factors in observable and unobservable variables.

**Weaknesses:**

I recommend to reject this paper, because I simply fail to understand it. I am familiar enough with the original VIN approach, but without reading the QMDP-Net paper (which is quite old by now) I doubt many will be able to understand this paper. What is needed here is a complete rewrite that explains (in equations) how the kernels are learned, how the belief is updated, and how the resulting VIN is actually solving the POMDP. In particular the main innovation, the pairwise approach, must be explained much more for the presented equations to make sense. I tried to read Section 3 a couple of times, but I am still unsure what is computed here, why it is computed, and how this is supposed to solve a POMDP.

If other reviewers disagree with this statement, I am happy to be convinced otherwise, but I believe an ICLR paper should be accessible even to non-expert of a field like this.

**Detailed Comments**
- To solve the induced belief-MDP, one needs to do value iteration over the space of all possible belief distributions. I do not understand how this can be achieved in VIN, which only seems to work over discrete state spaces (beliefs are continuous).
- The term uncertainty is ambiguously used: I believe you mean partial observability, and sometimes noisy observations, but uncertainty is more often associated with stochastic environments (aleatoric), or incomplete sampling (epistemic).
- Some symbols are never or insufficiently defined, like $Q$ in Equation 5. This extends to fairly important concepts as the belief distribution $b(s,s')=b(s)b(s')$, where it is never defined how these $b(s)$ are updated or why they are independent from each other.
- The notation often changes during the text. For example, the reward function is defined (and first used) as $R(s,a)$, but then later used as $R(s)$ or $R(s,a,z)$ without defining these terms formally.
- Equation 1 defines whether $s$ and $s'$ are *distinguishable*, but contains a sum over $s$ and $s'$, which does not seem to make any sense.
- It is unclear to me how the values $V(s, s')$ and $V(s_v, s_h, s'_h)$ are actually represented in the architecture.
- The experiments are missing baselines, i.e., other approaches to solve POMDPs. Just comparing to QMDP-Net is not enough for a top-tier conference.

**Questions:**

- Why should the actions selected in Equation 5 solve a given POMDP?
- What are the standard deviations in the tables over? Did you train the kernels multiple times and this is the STD over the random seeds? Is this about multiple runs of one set of kernels with noisy observations?

---

> ### Author Response · Authors · 2025-12-04
>
> We are disappointed to hear that our main innovation and its implementation were not clearly communicated. We have edited the main body of the paper to address clarity issues. Specifically:
>
> Comment 1- **Value Iteration over continuous belief space:** The reviewer is correct in identifying that beliefs are continuous while our VIN works on a discrete belief space. However, as the pairwise heuristic dictates, in addition to only iterating over pairs of discrete states, we also assume that the belief in each state is 0.5 for all pairs. This is done in order to further reduce the size of the belief space, reducing the computational complexity of our solution. This yields a discrete belief space for the value computation performed by VI$^2$N. After the value computation is performed with discretized states, the values of pairs are weighted based on the full continuous belief space.
>
> Comment 2 - **Definition of uncertainty:**  We have updated our introduction to emphasize the definition of uncertainty in our context – namely, the partial observability that is modeled by POMDPs.
>
> Comments 3–6 - **Updated Notation and clarifications:**
> The reviewer remarks that $Q$ is not defined in equation 5. $Q$ is a very common notation for the expected value of a state, action pair in reinforcement learning. We have updated the background section defining MDPs to reiterate the definition of this equation.
>
> Additionally, the reviewer states that the definition of belief and the belief update mechanism is not sufficiently explained. However, we introduce belief in Section 2 (background), under our definition of Partially Observable Markov Decision Processes.
>
> Regarding the confusion for the notation of Reward: $R(s, a, s’)$ is the classic reward notation used for MDPs. We simplified that to $R(s, a)$, assuming a fully deterministic action, but have changed it to $R(s, a, s’)$ at the reviewer’s recommendation to avoid confusion. The next representation, $R(s, a, z, s’)$, is only used in the Partially Observable case where the observation may impact the reward obtained. Again, we have updated this to $R(s, a, z, s’)$ for consistency. Finally, throughout our proofs in Section 3 we consistently use R(s) to denote reward for brevity, as our reward function is uniform across actions and future states. We have added this clarification to the beginning of Section 3.
>
> Finally, we clarified the calculation of $V(s)$ and $V(s, s’)$, and further refined the explanation of the POMDP solver and action selection.

---

### Official Review · Reviewer_6LxJ · 2025-10-30

**Soundness:** 3
**Presentation:** 4
**Contribution:** 3
**Rating:** 6
**Confidence:** 4

**Summary:**

The paper "Information-based Value Iteration Networks for Decision Making Under Uncertainty" proposes to combine Value-Iteration networks with Partially Observable MDPs (POMDPs) to account for uncertainty in the environment. To overcome the computational complexity of obtaining the optimal policy for POMDPs they adapt a solver that uses the pairwise heuristic that estimates the value $V(s,s')$ for states $s,s'\in S$. Notably, the authors argue that this heuristic is only necessary for the features that are uncertain, which can drastically decrease the amount of required computations. They provide empirical evaluation on two original gridworld datasets and show that their approach notably outperforms QMDP in environments with high uncertainty. Finally, they present "information maps" for each state that appear to make model decisions more interpretable than the value function.

**Strengths:**

The combination of the solid theoretical foundation of value iteration with MOMDPs to account for uncertainty in the environment is inspiring. The theory developed in this paper was very clearly stated and easy to follow, even without significant knowledge on partial observability or pairwise heuristics for such problems. Furthermore, the observable goal, unknown position environments were motivated by real-world applications (like robots with sonar sensors), justifying their relevance. The emperical results show the benefits of this approach for difficult navigation environments. On top of that, the analysis highlights that both reward exploitation and resolving uncertainty are essential for the model's success. Finally, the obtained information maps provide an impressive insight into the agent's capabilities.

**Weaknesses:**

As mentioned in the discussion, the lack of evaluation beyond 2D navigation tasks is unfortunate, since it would help to understand the algorithms capabilities in other reinforcement learning domains. Specifically, you state that you successfully tested noisy environments; it would therefore have been interesting to see quantitative results for these experiments, since they are especially relevant for real-world applications. Furthermore, though QMDP-Net performs better than unconstrained networks, a comparison to at least one state-of-the-art baseline not specifically built for partial observability would have been beneficial.

So, the main issues are:
- Limited experiments on relatively toy 2d environments
- Lack of comparison against other established methods

**Questions:**

I am unsure how the threshhold $\lambda$ has to be selected by a domain expert and why it should be close to 1. The reasoning behind this choice would be an interesting note to fully grasp the presented approach.

---

> ### Author Response · Authors · 2025-12-04
>
> We thank you for your review and hope to clear up some of your questions with our response.
>
> **Weaknesses:** We address the two weaknesses identified in our Author Comment above.
>
> Q1- **𝜆 selection:** The parameter 𝜆 represents a “distinguishability threshold”, which can be explained as how distinct a pair of observations must be to be considered distinguishing observations. In a noise-free environment, a pair of observations can only prove distinguishable if they do not have overlapping distributions, meaning 𝜆 would be 1. In noisy environments, observations can sometimes have overlapping distributions, yielding a 𝜆 slightly below 1. Both of our tasks utilize noise-free observations, and therefore, we use a 𝜆 of 1.

---

### Official Review · Reviewer_wGbv · 2025-11-07

**Soundness:** 1
**Presentation:** 3
**Contribution:** 2
**Rating:** 2
**Confidence:** 4

**Summary:**

This paper introduces new a differentiable planning architecture ($VI^2N$) for decision making under uncertainty in partially observable environments. The method extends Value Iteration Networks (VINs) by integrating a pairwise heuristic mechanism (from prior works) that aims to explicitly model and reduce uncertainty before exploiting rewards. The authors also proposed how $VI^2N$ can leverage Mixed Observable MDP (MOMDP) factorisation to reduce computational complexity and improve scalability. They then present various experiments in several simple grid-world domains with different degrees of uncertainty (covering cases where the agent’s position or the goal’s position is unknown) and shows that $VI^2N$ outperforms QMDP-Net (the most directly related prior work) in success rate. An ablation study on the recurrence depth of $VI^2N$ further highlights the importance of the model’s planning horizon.

**Strengths:**

* **Significance and originality:** The paper tackles an important limitation of previous value iteration-based networks (VINs, QMDP-Nets) by explicitly integrating information-gathering behavior into planning under uncertainty. This is highly relevant for domains such as robotic navigation and grid-world planning tasks.
* **Experimental coverage:** The study provides experiments in several partially observable gridworld settings, including tasks with unknown agent position / known goal and known agent position / unknown goal, capturing a different types of uncertainty.
* **Comparative results:** The paper demonstrates consistent and often substantial improvement over QMDP-Net, the closest relevant baseline, showing that $VI^2N$ can perform better in environments with significant perceptual ambiguity.
* **Insightful ablation:** The analysis on the number of recurrences in the VI and $VI^2$ modules reveals how performance depends on planning depth, providing useful interpretability and model understanding.
* **Clarity and theoretical grounding:** The exposition of the pairwise heuristic is conceptually clear and builds logically on established literature in POMDPs.

**Weaknesses:**

* **Limited generality:** Like QMDP-Net and other VIN-based models, $VI^2N$ assumes a discrete environment with spatially invariant transition kernels and discrete actions, which restricts applicability to continuous or large-scale real-world domains.
* **Figure clarity:** The main architecture diagram (Figure 1) is dense and difficult to interpret, lacking a proper legend and a lot of missing arrow direction indicators, which makes following data flow challenging.
* **Restricted evaluation scale:** Experiments are limited to small binary grid-worlds, which makes unclear how it will scale to larger maps or continuous domains (e.g., those used in the prior VIN or QMDP-Net works).
* **Outdated baselines:** The authors only compare against QMDP-Net, which is a fairly old baseline (Karkus et al., 2017). They justify this by saying that the QMDP-Net paper showed that they perform significantly better than unconstrained networks. However there are several more recent POMDP baselines other than RNNs/LSTMs and behavior cloning (which is what Karkus et al. compared against), like transformers (e.g Decision transformers, TrXL, etc). Additionally, given that the experiments are all trained offline using expert trajectories, there are several more recent offline RL baselines (like CQL, IQL, etc).
* **Weak evidence for claims:** It looks like the authors did not average their results across several training runs (or the number of seeds used is not reported). Hence it is unclear if the results are significant (let alone statistically significant). There are also no plots/results to validate the claim that $VI^2N$ focuses on reducing uncertainty *before* exploiting rewards. Finally, it is unclear if QMDP-Net also got/used the same factorised representation in Task 2 with the fully observable agent’s location (so the belief should similarly be only over unobserved variables).
* **Little ablations and analysis:** Only one ablation (recurrence depth) is provided. No analysis of other hyperparameters and model failure modes is given. For example it is unclear how performance varies with $\lambda$, and what tasks/situations are problematic for $VI^2N$ due to the changed bellman equation (Equation 4).
* **Lack of robustness tests:** Although the architecture is designed for high uncertainty, the paper does not test performance under systematic increased stochasticity. E.g.
  - Increasing action slip probabilities
  - Noisy observation models like the classical “noisy TV” scenarios where observations become uninformative (e.g. where transitions into a specific grid position gives uniformly random observations).
* **Scope of generalisation:** All experiments use offline expert demonstrations. Hence it is unclear how the method will be affected by non-expert demonstrations or the online RL settings.

Some of these limitations are acknowledged in the last section (Discussion), including a couple additional ones. However given the severe lack of analysis as mentioned above (and no theory), the authors really should have included some of the experiments they leave to future works.

**Questions:**

Please see the weaknesses above. Mainly:

1. **Baselines:** Why were no more recent POMDP or offline RL baselines included? Could you compare against transformer-based or uncertainty-aware architectures/algorithms (like CQL) to contextualise performance?
2. **Scalability:** How does $VI^2N$ scale computationally and in performance for larger or continuous environments? Have you tested on larger grid-worlds or 3D navigation tasks as suggested in discussion, since you claim it is easily doable?
3. **Statistical significance:** How many training seeds were used? Could you report mean ± std over multiple runs to assess (or at least illustrate) the significance of the improvements?
4. **Robustness to noise:** How would $VI^2N$ perform under increased stochasticity (e.g., transition noise or noisy observations)? Does the network maintain its advantage over QMDP-Net in such cases? Does the plots of informative areas through the value function of pairs change as one would expect?
5. **Ablation depth:** Beyond recurrence count, could you ablate other design elements (e.g., factorisation, pairwise distinguishability threshold λ) to clarify what drives improvements and the corresponding tradeoffs? Are there failure cases as a result of the way the rewards and Bellman equations are modified (or can the authors prove they maintain optimality)? In general could the authors analyse failure cases?
6. **Interpretability consistency:** Are the “informative area” maps consistent across runs and different environment structures, or do they vary significantly depending on training initialisation?
7. **Generalisation to non-expert settings:** Given that experiments rely on expert trajectories, how would $VI^2N$ perform under non-expert trajectories?

---

> ### Author Response · Authors · 2025-12-04
>
> We thank you for your constructive criticism and questions. Please find our response to issues raised in Q1, Q2, and Q3 as our Author Comment above. We further address Q4, Q6 and Q7 in the following:
>
> Q4- **Robustness to noise:** We have tested our model in stochastic environments and found that the results were quite similar to those in noise-free environments. We reason that this is because our models are still able to learn the transition function despite the presence of noise and to recalculate the belief from observations. Our stochastic test case consists of the following transition probabilities for each action: p(intended direction)=0.6, p(left) = 0.1, p(right) = 0.1, p(stay) = 0.2. We have updated our final submission to include these results in the appendix.
>
> Q6 - **Interpretability consistency:** The reviewer questions the consistency of our claim that our model values "informative” states consistently across different environment initializations. In section 4 (results), we demonstrate that our model visits the landmark state far more frequently than QMDP-net, visiting it an average of 96% of the time, whereas QMDP-net only visits it an average of 34% of the time. This consistent pattern of visiting the landmark is a direct consequence of the informative landmark state being valued higher than other states, as the model’s policy is a direct function of the value representation.
>
> Q7 - **Generalization to non-expert settings:** When training our model on non-expert trajectories, the network cannot learn meaningful value associations and therefore performs poorly on the data.  For example, we tested our model using a QMDP solver agent as the “expert” demonstrator. Since QMDP has a high failure rate at solving the problem (95% failure rate), this also yields a very low success rate in our model performance (3 %)
>
> Additionally, we follow up on an unaddressed concern raised in W5 - **Weak evidence for claims:**
>
> The reviewer brings up uncertainty about whether QMDP-net also gets a factorized representation of belief. While QMDP-net receives the same belief vector as input – ie. the cartesian location of the agent is fully observable while the true goal state is not – QMDP-net is not able to utilize a factorized representation of this belief due to its structure, which is an advantage that VI^2N has over QMDP-net.

---

### Author Response · Authors · 2025-12-04
**Response to general concerns**

We thank all the reviewers for their insightful feedback. While we answer each review separately as well, there are a few points that were raised by multiple reviewers, which we address first:

1- **Performance (success rate) report**: There was some confusion about our performance reports. All reported success rates across methods and tables are based on 10 random initializations of the weights (different seeds). Moreover, we reported the whole range of these 10 runs. For example, “57 $\pm$ 4%” means the lowest success rate was 53% and the highest was 61% across 10 random initializations. This shows that, in many cases, the highest QMDP-net success rate was still lower than the lowest VI$^2$N success rate. We can change our reports to the mean and standard deviation across runs. Still, given the slight variability in success rates across methods (mostly around 2%, with a max of 5%), reporting the full range would be more useful.

2- **Limited experiments**: Our experimental results are based on two sets of 2D navigation tasks. While these two sets are both about finding a goal, their structures, especially in the Markovian framework (e.g., observation function), are fundamentally different. Not only is the type of uncertainty different in these two problem sets (uncertainty about the position of the goal or the agent itself), but also observation kernels differ, where one is based on the agent’s surroundings, and the other is based on a specific landmark revealing the position of the goal that could be far away. Therefore, the observation function can be implemented by using a small CNN-like kernel in the first problem. In the second problem, the observation function has a similar structure to the reward function, representing the whole state space (N * N map).

Moreover, an important advantage of our method, as appreciated by some reviewers, is its interpretability. This feature is best conveyed in 2D navigation tasks where high-reward areas and informative areas can be easily visualized.  For example, Figure 3b shows states with high value concentrated near the true goal and landmark states.

In addition, each of our problem sets contains multiple types of environments that examine the performance of methods in a systematic way. For example, in our first set, we looked at structures with different levels of available information for the agent to figure out its own position. Our experimental setup is designed to have a comparison between methods beyond mere numbers. As mentioned in the discussion, designing problems with varying levels of complexity is very difficult, and some test environments used in the literature are not even challenging for the classic QMDP algorithm. Notably, common test benchmarks such as gym and Atari are not applicable to our work as they do not contain uncertainty/partial observability. In fact, decision making under uncertainty is highly underresearched due to its complexity, resulting in very limited problem-sets and also methods. Finally, it's worth mentioning that one of the limitations of VIN-based networks is their dependence on spatially invariant transition (and observation) functions. In practice, this is not a major barrier, especially in fields like robotics, but toy problems with unstructured functions cannot be solved by these networks.

As requested by the reviewers, we have added experiments that test our model in a larger environment and in a stochastic environment.

3- **Other Baselines**: QMDP-net is a relatively old paper, given the pace of the field in general. However, as mentioned above, as decision-making under uncertainty is an under-researched field, QMDP-net is still the best deep network for partially observable tasks. Other more recently developed models, such as Transformer-based models, have only been tested on fully observable tasks. In fact, we tested Decision Transformer (Chen 2021) on our second task (uncertain goal position). We find that the Decision Transformer performs poorly on the task, achieving a success rate of 3%. This very low performance is consistent with failures of other RL networks in partially observable environments, even though they were very successful in fully observable environments, as shown in (Ni et al 2022).

---

### Author Response · Authors · 2025-12-04
**Summary of Additional experiments**

In addition to addressing reviewers’ concerns, we have edited our submission for clarity, as requested, especially in section 3, and have taken time to define the concepts introduced in our paper more clearly. Moreover, we added several experiments to problem set 2, which is more challenging, as explained in the comments and the edited version of the paper.

In summary, our additional experiments are:
- Larger environments
- Noisy environments (same as the original problem set but with noise)
- Decision Transformer

---

### Meta-Review · Area_Chair_biF5 · 2025-12-17

**Summary:**

This paper proposes VI²N (Value Iteration with Value of Information Networks), a differentiable planning architecture for decision making under partial observability. The central idea is to incorporate information-gathering directly into the planning process by embedding the classical pairwise heuristic for POMDPs into a value iteration network. By explicitly planning over pairs of states, the method accounts for uncertainty reduction rather than relying on the QMDP assumption that uncertainty disappears after one step. The paper further extends this approach to mixed-observability settings (MOMDPs), where factorization of the state space significantly reduces computational complexity. The approach is evaluated on a range of grid-based navigation tasks with different types and degrees of perceptual ambiguity, where it consistently outperforms QMDP-Net. An additional strength of the method is interpretability: beyond standard value maps, VI²N produces information maps that highlight states important for resolving uncertainty.

Overall, this paper makes a principled and non-trivial contribution to differentiable planning under partial observability by explicitly integrating information value into the planning process. While the scope of the empirical evaluation is limited and scalability beyond structured domains remains an open question, these limitations are clearly acknowledged and do not detract from the conceptual and technical merit of the work. I recommend acceptance as a solid, focused contribution to the literature on planning-based reinforcement learning under uncertainty. I therefore recommend acceptance.

**Reviewer Concerns:**

The reviewers raised several valid concerns, most of which relate to scope and evaluation rather than correctness or conceptual soundness. A recurring point was the limited experimental domain, as all experiments are conducted in 2D navigation tasks. While this limits the generality of the empirical validation, the chosen domains are well aligned with the paper’s goals: they isolate partial observability, information gathering, and long-horizon planning in a controlled and interpretable manner. The authors explicitly acknowledge this limitation and justify the choice by noting both the lack of established benchmarks for deep POMDP planning and the difficulty of constructing non-trivial uncertainty-driven tasks in other domains. Importantly, the experiments are not ad hoc; they are systematically designed to stress precisely the failure modes of QMDP-based approaches.

Another common concern was the lack of modern baselines, particularly transformer-based or offline RL methods. This is a reasonable request given current trends. The authors addressed this by clarifying that most modern architectures have been evaluated primarily in fully observable settings, and by adding an explicit comparison with a Decision Transformer on the uncertain-goal task, where it performs very poorly. While the baseline coverage remains limited, the paper makes a reasonable case that QMDP-Net remains the most directly relevant comparison for differentiable planning under partial observability, and the added experiment helps contextualize performance relative to more recent methods.

Reviewers also questioned scalability and computational complexity, especially given the quadratic nature of pairwise planning. The paper is clear that unfactorized pairwise planning is expensive, but it also shows that MOMDP factorization dramatically reduces complexity in structured settings. The authors further clarified computational tradeoffs in the revised manuscript and emphasized that the approach is intended for domains with exploitable structure rather than as a general-purpose POMDP solver.

Several reviewers raised concerns about statistical reporting, robustness, and ablations. The authors clarified that results are based on multiple random initializations and explained their reporting format. In response to robustness concerns, they added experiments with stochastic transitions, showing that the method maintains its advantage over QMDP-Net. While additional ablations could strengthen the paper further, the existing recurrence-depth analysis already demonstrates the necessity of both the reward-planning and information-planning components.

**Reviewer Scores:**

Reviewer 6LxJ already expressed a mildly positive assessment and indicated openness to either outcome. Given the authors’ clarifications regarding the distinguishability threshold, additional robustness experiments, and improved presentation, it is unlikely that this reviewer would lower their score. If anything, their score would likely remain unchanged or increase slightly, but still in the “borderline accept” range.

Reviewer bA3z raised concerns about limited evaluation, novelty, statistical reporting, and clarity of the training setup, while explicitly stating openness to changing the score if these issues were addressed. The authors’ responses directly addressed statistical reporting, clarified the training regime, tightened claims about domain diversity, and expanded explanations of novelty and computational tradeoffs. Based on the tone of the review and follow-up comments, it is reasonable to expect this reviewer would increase their score modestly, likely moving from marginally below threshold to marginally above threshold.

Reviewer wGbv identified many valid limitations but also acknowledged the conceptual grounding and empirical improvements over QMDP-Net. Several of this reviewer’s strongest concerns—statistical significance, robustness to noise, and missing baselines—were partially addressed through clarification and additional experiments. While this reviewer would likely still view the paper as limited in scope, their objections appear to be more about maturity and breadth than correctness. With full participation in discussion, their score would plausibly increase slightly, though likely remain on the cautious or weak-reject side rather than flipping fully to acceptance.

Reviewer SnV9 explicitly stated difficulty understanding the paper and expressed low confidence in their assessment. The authors substantially revised the presentation and clarified several of the points this reviewer found confusing, particularly regarding belief discretization and the role of the pairwise heuristic. However, given the reviewer’s own admission of unfamiliarity with the core technical background, it is unclear whether participation in discussion would have led to a meaningful score change. At best, the score might increase marginally due to improved clarity, but it is also plausible that it would remain unchanged.

In summary, the discussion and author responses are likely to have resulted in net upward movement in scores, particularly among reviewers who engaged substantively with the method. Remaining disagreement appears to stem from differences in expectations about scope and evaluation breadth rather than unresolved technical flaws.

---

### Decision · Program_Chairs · 2026-01-26

Accept (Poster)